# Effects of Online Problem-Based Learning to Increase Global Competencies for First-Year Undergraduate Students Majoring in Science and Engineering in Japan

**Eri Ota * and Rie Murakami-Suzuki** 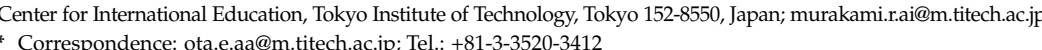

Center for International Education, Tokyo Institute of Technology, Tokyo 152-8550, Japan; murakami.r.ai@m.titech.ac.jp
* Correspondence: ota.e.aa@m.titech.ac.jp; Tel.: +81-3-3520-3412

**Abstract:** The purpose of this paper is to assess learning outcomes and the process of developing skill sets of online problem-based learning (PBL) for students majoring in science and engineering at a technical university in Japan. PBL course for first-year undergraduate students was organized with international students as teaching assistants (TAs) to find issues and solutions of the target countries. Due to the effects of COVID-19, the course was completely shifted online in the 2020 academic year. Topics selected by group members were all in line with sustainable development goals (SDGs). Three skill sets to be developed through this PBL course are global awareness, problem-solving and finding, and multicultural communication and understanding. A series of analyses on text mining and content analysis was conducted of essays and reports submitted by students registered for the course. This paper summarizes the structure and pedagogy of the course, research methods, research results and reasons for findings. Research results indicated that all three skill sets were well developed in students through this online PBL course by conducting a study of selected countries with group members, carefully listening to other groups' presentations in the class, conducting data analysis and online interviews, and communicating with TAs in English. Based on the findings, efforts to maintain quality education in conducting online PBL are also discussed.

**Keywords:** globalization; global awareness; problem finding and solving; multicultural understanding; online problem-based learning; international students

## 1. Introduction

One of the important roles for higher education institutions is to grow human resources contributing to the future well-being of society. Sustainable Development Goals (SDGs) set by the United Nations are often utilized as guidelines for the enhancement of the whole world [1]. Currently, many people—regardless of their ethnographical and economic backgrounds—are affected by global issues such as climate change, energy, and demographic change, the causes and solutions of which are complicatedly linked and related. In promoting sustainable development by tackling global issues, the involvement of various stakeholders is necessary. Many studies such as that by Chin et al. insist on the role of engineering education in promoting sustainable development [2–5]. Engineers can take active roles in making innovations in terms of technology development and understanding of mechanisms on natural phenomena. In order to come up with innovations, it is necessary to work with people with multicultural backgrounds, since diversity will bring different perspectives and new ideas [6–8]. Allan and Chisholm insist that engineers should develop global competencies such as embracement of ethics, presentation of empathy, critical thinking, development of culturally appropriate relationship, utilization of information and communication technology (ICT), ability to work within inter and transdisciplinary systems, and interacting interpersonally under culturally/ethnically/linguistically different situations [6]. Reimers presents three dimensions of global competencies as a positive approach toward cultural differences, willingness to engage those differences,

ability to speak/understand/think in several foreign languages, as well as broad and deep knowledge of world history/geography/global aspects of issues [9]. According to Gao and Hite as well as Guerra, their research findings suggest that many skills necessary to equip with global competencies such as the effective use of technology, communication, problem-solving, and global mindset can be developed through problem-based learning (PBL) [10,11].

The authors have been conducting an in-class PBL course titled "Introductory Course for Global Scientists and Engineers" for first-year undergraduate students majoring in science and engineering for eight years at a technical university named Tokyo Institute of Technology, or Tokyo Tech for short. The students' learning objective of this PBL course is to equip with basic global competencies namely global awareness, multicultural understanding, critical thinking and problem solving, and English communication by conducting group work with international students as teaching assistants (TAs). Without exception, similar to most universities in many parts of the world and due to the worldwide spread of the new virus COVID-19, the course was forced to shift from face-to-face to completely online in the 2020 academic year, based on the university policy to refrain from organizing in-class courses to avoid infection. Thus, the term provided by Hodges et al. of "Emergency Remote Teaching (ERT)" [12] was implemented, which was the primary reason for shifting the course to being organized online.

Having been affected by COVID-19, global education in higher institutions needs to proceed with a big transition. Specifically, in order to implement global education under the new paradigm, it is necessary to change the pedagogy, keeping in mind that one of higher institutions' missions is to increase global awareness and contribute to students' future career formation. Many publications reveal the pros and cons of online PBL courses in comparison to traditional classroom-style PBL, before and after COVID-19. During the COVID-19 pandemic, even though it was ERT, empirical studies related to effective online learning have been shared, which could be utilized as new forms and opportunities of global education. For example, ERT implementation at the School of Telecommunication Engineering at the University of Spain, the research by Iglesias-Pradas et al. reported that students' academic performance was well kept [13]. In the case of a university in Latvia, it was reported that students and faculties were well adapted to online courses, which can be classified as ERT, and see the new prospects as new ways of teaching in the long run [14]. Şendağ and Odabaşi discuss that there are no particular differences in learning outcomes between online and face-to-face teaching [15]. The authors also reported the same findings that, based on the result of "Course Survey of Study Effectiveness," the standardized survey conducted at Tokyo Tech, it was assumed that there were no particular differences in overall learning outcomes of students for the 2020 academic year from previous years. After successfully conducting online PBL, the authors confirm that there are many implications of shifting the teaching style and utilizing digital devices regardless of COVID-19.

With the need for engineers to be equipped with global competencies, many publications related to face-to-face PBL present findings on effective pedagogies and challenges in developing necessary skills. Articles related to online PBL courses mainly focus on ensuring learning outcomes. Many studies related to the utilization of online media to promote multicultural understanding have also been introduced. On the other hand, few studies have been conducted for online PBL focusing on actual pedagogical processes within engineering education. Taking into consideration the needs of pedagogical research, this paper aims to elucidate the learning process of online PBL for students to equip engineers with basic global competencies, which are global awareness, problem-solving and finding, and multicultural understanding and communication. This research therefore tries to examine effective pedagogies for online PBL to see its unique characteristics in comparison to face-to-face PBL. In so doing, this paper (1) reviews the existing literature for engineering students to equip themselves with global competencies and the effectiveness of online global education, (2) briefs the course content and pedagogy of online PBL conducted at

Tokyo Tech in the 2020 academic year, (3) explains research methods, (4) discusses research results and reasons for finding, and finally (5) presents a discussion and conclusion.

## 2. Background

### 2.1. Global Competencies Development in Engineering Education

Of many pedagogies to prepare engineering students to work in the international arena, PBL is considered to be effective as it forms small groups of students working to learn the course content within the framework of a realistic issue [16–19]. It is expected that through PBL, students will be brought up with skills related to effective communication, critical thinking and analysis, and lifelong learning development [20–22]. Studies also suggest that using mobile technologies for culturally responsive PBL is a powerful and unique way to prepare students for the four Cs: critical thinking, communication, collaboration, and creativity [19]. In delivering education related to sustainability, Guerra stated the effectiveness of PBL on cross groups in terms of expertise, department, and disciplines [23]. Gabdulchakova et al. elaborated that multicultural awareness can be developed at the teaching level by utilizing multiple technologies such as communicative technology, critical thinking technology, case study technology, problem-based learning and moderation technology, and an expert problem seminar. The article further discusses that by utilizing these technologies, students can progress to understanding the problem and determining the ways for solving the problem [24]. Furthermore, Boothe et al. stated that PBL is an important means to deliver STEM courses that enroll students who speak different languages, especially with insufficient English ability [25]. Furthermore, according to Du and Kolmos as well as Voronchenko et al., multicultural PBL is beneficial because it would motivate students to further increase their levels of interest in topics, initiate conceptual change, bring cognitive engagement, and develop respect and acceptance, as well as tolerance to others [18,26].

In summary, in order to develop global engineers capable of solving complicated issues, along with their field of expertise, engineers of the 21st century need to be equipped with soft skills such as multicultural understanding and communication, critical thinking, and global awareness. Tokyo Tech, a leading higher institution in Japan that has a long history of growing human resources in the field of technology development and application, has set students' global competencies development as one of the core academic policies since 2013. Many of the teaching methods, such as utilizing ICT tools, collaboration with partner universities in different countries, delivering lectures and workshops by experts, have been applied. On the other hand, prior to COVID-19, many courses were held on-site, either in the classroom or field, and both domestic and abroad.

### 2.2. Effectiveness of Online Educations

Prior to the COVID-19 incidence, many universities have implemented online PBL, and thus the effectiveness and potential of online education to enhance global competencies has been recognized. Brodie proved that a fully online PBL course can be successfully delivered to engineering students via distance education in which students work in a virtual mode by conducting online team meetings to solve complex, real-world engineering problems [27]. Munoz-Escalona et al. demonstrated the effects of collaborative online international learning (COIL) as one of the effective pedagogies to stimulate students' international opportunities in the future, through which students from institutions in different countries compare each other's knowledge about selected topics [28]. A similar statement was shared by Zhang and Pearlman—that technology-enhanced COIL can help increase intercultural awareness and communicative competence and promote understanding of global issues in a cost-effective way [29]. Merryfield raised the point that for online teaching to promote multicultural understanding, it is compelling for students in taking risks, sharing personal experiences as well as real intentions, some actions of which can be limited in face-to-face communication. The paper further discusses that in order to increase global awareness, an understanding of culture, power balance, and worldview is necessary [30].

Discussing a similar concept from the other side, Chametzky highlighted that increasing learners' motives can be realized online when real-world problems are included in content and collaborative work is implemented [31]. Furthermore, research conducted by Madleňák et al. proved that well-planned flipped learning along with proper learning material provision can ensure an increase in learners' motivation, knowledge acquisition, confidence, and satisfaction. In so doing, the paper insisted that instructor–students communication and multiple approaches for checking content acquisition are important [32]. Research conducted by Şendağ and Ferhan Odabaşi concluded that students' critical thinking skills can be increased more so through online PBL than an instructor-led online course [15]. Wojenski indicated that the utilization of online tools could be a supplement or replace actual international experience by overcoming some limitations to accessibility in terms of the cost and conflict with other academic schedule [33]. Bruhn also mentioned that the internationalization of education includes an introduction of global learning components to the curriculum held domestically, without physically crossing the national border; thus, teaching delivery is possible on-campus or through online education [34].

There are a few cases for online PBL held in collaboration with universities from different countries. One example is taking ocean pollution as a shared theme between China and the United States, and it has been proven that global awareness has been promoted among students [10]. Another example is online multicultural PBL, developed by Tokyo Tech in Japan and Chulalongkorn University in Thailand. In articles, Ota et al. have shown that a combination of face-to-face visits of two countries and online PBL is a very effective pedagogy in increasing global awareness and can promote skills such as critical thinking and multicultural teamwork [35,36].

While many pieces of literature confirm the effectiveness of online PBL, there are also some studies which indicate limitations and difficulties in conducting online PBL. For example, Savin-Baden delivers some concerns on online PBL about students' comfortability in communication [37]. Furthermore, research by Foo et al. shows lower scores on skills acquisition through online PBL compared to face-to-face PBL. According to the article, it was reported that through online PBL, students' growth was low in skills assumed to be developed through PBL, such as group work skills, including participation, communication, and preparation. Some of the listed reasons are phycological barriers as well as the minimum time for adaptation to new method during COVID-19 [38]. A case in a university in Pakistan experienced somewhat unfavorable results with remote teaching. Adnan reported that with both limited technical and financial capacity, many students in Pakistan felt uncomfortable with online course delivery and preferred traditional face-to-face learning [39].

As for learners' satisfaction and maintaining educational quality through online learning, studies exist in favor of online learning as well as mention some challenges. According to Zhan and Mei, there was no significant difference between face-to-face and online students with regard to the effect of academic self-concept on learning achievement and satisfaction. On the other hand, the article indicates that students joining online courses are in greater need of higher-level social presence and supports for social interaction than students taking face-to-face courses, which has a strong effect on their learning achievement and satisfaction [40]. A similar finding is presented by Zhou and Zhang—that interactions between the teacher and students were not enough during the implementation of an online course during COVID-19, which affected students' motivation [41]. In the case of Indonesia, Junus et al. reported that teachers experienced a difficult time in preparing materials, course delivery, and students' evaluation [42]. The need for greater attention by students and the feel of togetherness among students and teachers are raised in many publications, such as articles by Tang et al. and Jamalpur et al. [43,44]. Adeloyin and Soykan conceive COVID-19 as both a challenge and opportunity to progress online education. In the article, challenges include the need for proper planning for implementing effective pedagogy and technology for online learning, and one of the opportunities include increased or being "forced" to be equipped with some technological implementation to

deliver and receive online courses [45]. In articles related to the effectiveness of online PBL, a few tips for technological enhancement to ensure learners' knowledge acquisition and motivation is also discussed. For example, in an article by Hrastinski, along with an introduction to characteristics and benefits of asynchronous and synchronous online learning, it is mentioned that for online learning to be effective and efficient, instructors, organizations, and institutions must have a comprehensive understanding of the benefits and limitations [46]. Furthermore, the needs of a carefully planned online course have been insisted on by Gottipati, who also stated that best practices include interaction through the chat function [47].

From the literature review above, it is summarized that in implementing online PBL in engineering education to promote global competencies, some barriers and difficulties certainly exist. On the other hand, if carefully planned online PBL can be implemented by utilizing different types of learning materials, ICT tools, and communication channels, students can be well equipped with global competencies such as global mindset, critical thinking skills, and communication skills to work with people from different backgrounds. In doing so, instructors should pay full attention to students' learning and be well-prepared to deliver online PBL so that students' knowledge acquisition and skill set development are realized. The following section discusses the pedagogy and structure of online PBL implemented at Tokyo Tech.

## 3. Course Structure and Pedagogy

At Tokyo Tech, a PBL course titled "Global Scientists and Engineers Course" has been offered since 2013. The course is placed within an introductory curriculum to enhance global competencies for first-year undergraduate students. The course objectives are set for students to (1) understand global issues in targeted countries; (2) consider the relationship between Japan and the selected country and the role and contribution of Japan; (3) realize the importance of task sharing and effective communication; (4) develop qualitative and quantitative information collection and analysis skills; and (5) develop problem finding and solving skills. In summary, this course aims for students to increase global awareness, and equip them with problem finding and solving skills as well as multicultural understanding and communication. Through this course, students will explore the role of Japan in the era of globalization.

One of the unique characteristics of this PBL course is that students engage in PBL with international students serving as TAs for each group and learn about issues in the countries of TAs' origins. Through communication with international students from different countries, students will learn the current issues in other countries along with first-hand information from people from target countries without actually visiting them.

After an introductory lecture to understand global trends in relation to everyday life, students are asked to choose five preferred countries on which they would like to conduct research. Instructors then finalize the group member lists according to students' preferences. In the 2020 academic year, 45 groups consisting of four to five members were formed, in which international students were assigned as TAs. They were from 12 countries, namely Austria, Bangladesh, China, Ecuador, Egypt, India, Indonesia, Malaysia, Mexico, Philippines, Thailand, and Vietnam. The course was divided into four classes across two different days. Two instructors conducted courses simultaneously with the same materials and pedagogy. TAs' roles were to (1) provide comments and advice from the perspective of international students; (2) facilitate group work when necessary; and (3) coordinate online interviews with the person(s) in the target country. TAs submitted the group work evaluation for each group member for performance reviews.

Figure 1 shows the course structure and flow. The course was composed of an orientation by the instructor, video lectures by experts involved in official development assistance (ODA) in Japan, group work with TAs, topic and final presentations, and submission of final group reports and essays. Each international student from different countries participated in the group work as a TA and provided basic information and information related

to issues in his/her county of origin. Based on the information provided by TAs, students chose the topics that they wanted to work on, then conducted group work and studied issues in their target countries.

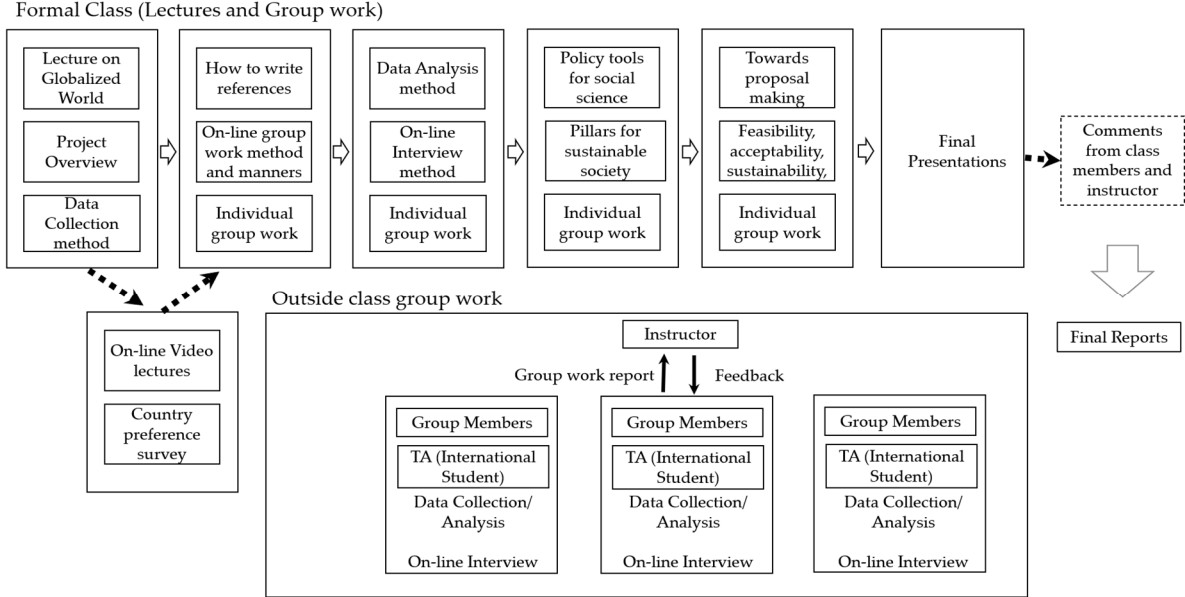

**Figure 1.** Course structure and flow.

Each class is composed of a lecture and group work. Lectures are conducted to assist students to understand the knowledge as well as the process necessary to find problems and develop a proposal for selected topics. Contents of the lectures are (1) group work project and research methods, (2) proper ways to make citations and references, (3) data collection, comparison, and analysis methods, (4) online interview methods, and (5) policy tools for problem-solving of social issues. Table 1 shows the structure of the final presentation and group report. Prior to submitting the final assignments, each group makes a topic presentation and submits results of data comparison and analysis as assignments.

**Table 1.** Structure of final presentation and group report.

| Chapter | Content |
|---|---|
| 1. Introduction | 1.1. Data comparison between selected country and Japan<br>Basic country data and data related to the selected topic, Comparative analysis with Japan (Utilize Data Comparison Sheet) |
| | 1.2. Research Method and Procedure<br>Method and procedure employed to conduct research |
| | 1.3. General description of the research topic<br>Reasons for selecting the topic, background of a key issue associated with the topic |
| 2. Current Situation around the Topic | Socioeconomic background, current policy, measures, and their evaluation |
| 3. Proposal for the Topic | Technology, policy, any other efforts to be applied |
| | Background of the proposal, ① Relevance, ② Coherence, ③ Effectiveness, ④ Impact, ⑤ Efficiency, ⑥ Sustainability |
| 4. Discussion | Possibility for collaboration with Japan for the solution/improvement of the key issue associated with the topic (including the relationship between Japan and the selected country) |

During the first part of the course, students watched two videos outside of the formal class. Students answered five questions and submitted a report regarding their learning of project implementation. One video is mandatory; students then choose one video out of three to watch. The mandatory video is about problem and solution finding based on actual projects, and the other three videos introduce actual cases of problem and solution finding implemented as part of Japanese ODA. After a topic presentation by each group, instructors sent comments and guidelines to group members and TAs to guide further studies, such as suggested data and information collection items.

After deciding the topics to work on, students were asked to conduct data collection and analysis about their target countries and Japan. Students collected data on two countries (Japan and target country) in two different years, in particular data available for the most recent year and 30 years ago. The basic data to be collected are literacy rate, total population, labor population, urban population, GNI per capita, income inequality index, birth rate, and people with at least a secondary education. It is also recommended that students pick some items for additional data related to their selected topics in order to further elucidate the issues. After receiving the results of data collection, comparison, and analysis, instructors sent comments, feedback, and advice to each group about their findings for further studies. After data analysis, students would further conduct a literature review and report their findings at the formal class and discuss the progress with TAs and instructors.

An online interview coordinated by TAs was the first trial implemented for this PBL course in the 2020 academic year. One of the primary purposes was for students to show a realistic situation of the target countries when international mobility was almost prohibited due to the COVID-19 pandemic. After briefing about types of interviews, objectives, and general structure, along with the international students' assistance, students conducted online interviews to either know about the current situation of the selected topics or to receive advice for the proposal they were discussing. As it would be the first time for many of the first-year students to conduct interviews in English, instructors provided a basic template for the flow of the interviews.

Every week, each group submitted a group work report to inform instructors about their progress on, for example, points of discussion, decisions made, and any problems that occurred. Within one to two days, instructors would reply to group members with comments and suggestions.

At the time of final presentations, students were asked to make comments in a few sentences for each group's presentation. After collecting comments for each group from all students, sorted by instructors, they were shared back to each group along with instructors' feedback. Along with final group reports, students submitted essays about their learning across three categories, namely global awareness, problem finding and solving, and multicultural understanding and communication.

The primary tool for online PBL utilized in the 2020 academic year was Zoom. The educational account was provided to instructors, and thus the "regular meeting" function was utilized to maintain the same Zoom link from the beginning to the end of the course. After the plenary lecture by instructors, breakout rooms were created for each class to organize online group work, and TAs were assigned as co-hosts. Instructors then joined each breakout room to check the progress and consult with each group. The chat and voting functions were utilized during plenary lectures to promote the active engagement of students in the course. Specifically, the chat function was utilized for students to make comments and opinions and also ask questions, while the voting function was utilized for small fun quizzes about globalization.

The Learning Management System (LMS) developed based on Moodle at Tokyo Tech called T2SCHOLA was utilized as an online learning platform. T2SCHOLA was effectively utilized for the submission of assignments, sending reminders and information necessary to proceed during the course, storing learning materials such as information sources, manual for journal article database, how to make proper references, online interview template,

along with online video lectures and lecture archives for each of the lectures/instructions provided in each class.

## 4. Research Method

185 students that registered and completed the course were the targets for this research. As mentioned in the previous section, this online PBL course was divided into four classes across two different days. Two instructors conducted courses simultaneously with the same materials and pedagogy. It is thus assumed and also proved from the previously mentioned "Course Survey of Study Effectiveness" that there were no remarkable differences in the results of learning outcome across the four classes.

In order to assess learning outcomes, three kinds of text mining by using the application called KH Coder on the essays submitted by each student and content analysis of final reports were conducted. For the essays, each student described their learning outcome and processes in approximately 900 Japanese characters, which is equivalent to 500 words in English, for the three following skill categories: (1) global awareness, (2) problem finding and solving, and (3) multicultural understanding and teamwork. Three kinds of text mining were conducted, as follows. First, an assessment of the high frequency of words that appeared in the essays was organized. The top 20 words that appeared in essays in each category were listed. Second, a co-occurrence network analysis was conducted to further understand the learning outcomes of students. Finally, in order to further understand the contents of the description and appearance of the words, a concordance analysis was conducted for the frequently utilized nouns in the top five most utilized words listed in the three skill categories and the results of co-occurrence analysis for some of the skill categories.

Furthermore, content analysis on final group reports was organized to elucidate the utilization of data and online interviews. Specifically, a selection of topics was categorized according to SDG themes to further assess students' interests in global issues. Furthermore, the inclusion/reflection of relevant data and results of online interviews within the finding of current situations and proposal-making on selected topics was examined.

## 5. Research Result

### 5.1. Frequency

Table 2 shows the top 20 words used frequently in the essays individually submitted by students after completing the group final presentations and reports. For the global awareness, the five most frequent words are "Japan," "problem," "think," "consider," and "country". It must also be noted that the words "research," "solve," "group," "contribute" were also utilized by many students. For problem finding/solving, "data," "solve," "propose," "consider," and "problem" are the five most frequent words that appeared in essays. Other words that appeared often are "research," "Japan," "collect," "problem," "information," "compare," "interview," "analysis," and "search". For multicultural understanding/teamwork, "group," "English," "think," "work" and "TA" are the five most frequent words that appeared through text mining. Other words utilized often include "member," "communication," "opinion," "active," "difficult," "contribute," "online," "Zoom," and "leader".

**Table 2.** Results of text mining of personal essays (Original data collected from submitted essays from students registered in "Introductory Course for Global Scientists and Engineers" in February 2021).

| Global Awareness | | Problem Finding/ Solving | | Multicultural Understanding/ Teamwork | |
|---|---|---|---|---|---|
| **Word** | **Frequency** | **Word** | **Frequency** | **Word** | **Frequency** |
| Japan | 482 | Data | 319 | Group | 402 |
| Problem | 411 | Solve | 253 | English | 285 |
| Think | 295 | Propose | 240 | Think | 282 |
| Consider | 281 | Consider | 220 | Work | 215 |
| Country | 254 | Problem | 215 | TA | 206 |
| Feel | 170 | Research | 159 | Myself | 182 |
| Research | 168 | Japan | 146 | Member | 174 |
| Solve | 153 | Collect | 138 | Communication | 167 |
| Myself | 150 | Think | 132 | Opinion | 142 |
| Group | 138 | Problem | 126 | Feel | 124 |
| This time | 127 | Information | 120 | Active | 115 |
| Develop | 99 | Compare | 118 | Talk | 109 |
| Presentation | 98 | Interview | 115 | Difficult | 105 |
| Contribute | 97 | Act | 98 | Active | 87 |
| Person | 95 | On site | 93 | Contribute | 87 |
| Many | 95 | Analysis | 78 | Online | 79 |
| Future | 92 | Person | 75 | Consider | 77 |
| Transportation | 91 | Topic | 70 | Zoom | 75 |
| Environment | 88 | Feel | 70 | Propose | 68 |
| Culture | 87 | Search | 67 | Leader | 65 |

*5.2. Result of Cooccurrence Analysis*

Figure 2 shows the results of co-occurrence analysis for essays on global awareness. The word "research" is closely connected with the words "problem," and "group". The word "country" is connected with the words "solve," "consider," "think," and "group". The word "group" has a close relationship with "presentation". Figure 3 shows the result of co-occurrence analysis for essays on problem finding and solving. The word "data" is closely linked with the words "problem," "consider," "think," "solve," and "proposal". A link was found between the words "Japan," "compare," and "collect". The word "interview" has a close link with the words "information" and "on-site". Figure 4 shows the result of co-occurrence analysis for essays on multicultural understanding and communication. The word "English" appeared in the center of cooccurrence analysis results, with a strong link with the words "TA," "communication," "group," "myself," and "think". The word "difficult" appeared in connection with "communication, "group," and "work. Two words, namely "contribute" and "positive," have a connection with the words "TA," "myself," and "English". The word "TA" has strong tie among words "member," "English," and "myself".

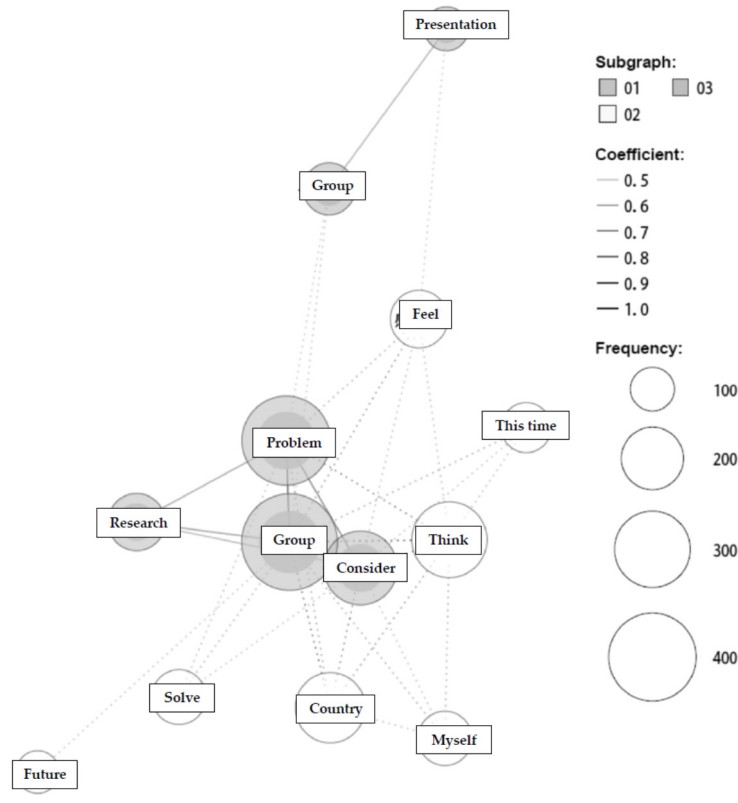

**Figure 2.** Results of co-occurrence analysis for essays on "global awareness." (Original data collected from submitted essays from students registered for "Introductory Course for Global Scientists and Engineers" in February 2021).

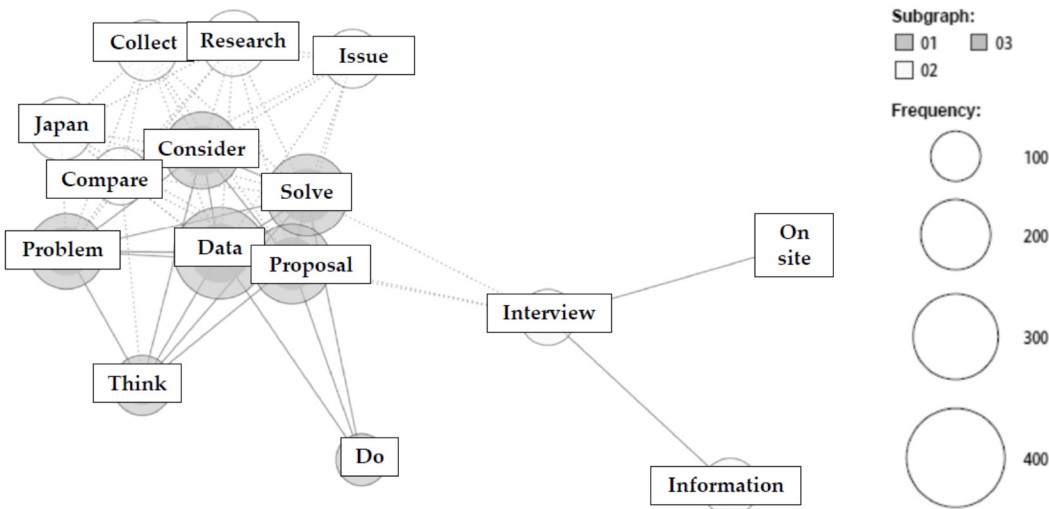

**Figure 3.** Results of co-occurrence analysis for essays on "problem finding and solving." (Original data collected from submitted essays from students registered for "Introductory Course for Global Scientists and Engineers" in February 2021).

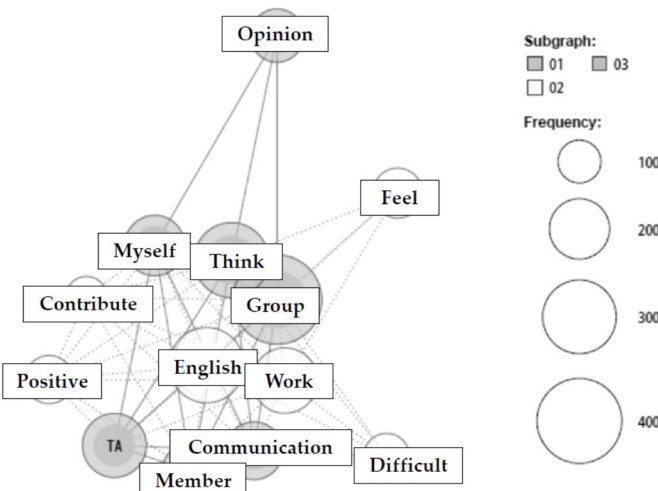

**Figure 4.** Results of co-occurrence analysis for essays on "multicultural communication". (Original data collected from submitted essays from students registered for "Introductory Course for Global Scientists and Engineers" in February 2021).

*5.3. Result of Concordance Analysis*

In order to further understand the contents of the description, a concordance analysis was conducted for the most frequently utilized nouns in the top five most utilized words listed across three categories. Furthermore, the same analysis was conducted for the links that stand out from other co-occurrence analyses, which are "group" and "presentation" in global awareness and "interview," "on-site," and "information" in problem finding and solving. Table 3 shows the context utilization of frequent words in each skill category.

For global awareness, three words, namely "Japan," "Problem," and "Country," were analyzed. The most frequently used word for global awareness "Japan" was utilized by 77 sentences in the context of comparing with other countries. In addition, the word "Japan" was utilized along with the word "country" 89 times, "problem" 39 times, "study" 24 times, "similarities and differences" 22 times, and "other countries" 21 times, respectively. Furthermore, the word "problem" was utilized in relation to other countries' problems in 61 contexts and students' own selected countries in 53 sentences. Another notable finding is that 59 students mentioned the word "problem" by referring to students' own field of expertise. In addition, the word "problem" was utilized along with the word "solve" 89 times, "environment" 27 times, and "society" 25 times, respectively. The word "country" was utilized in relation to the word "Japan" 89 times, in the context of selected countries 46 times, and countries in the world 46 times. The sentences which include "presentation" appeared 98 times, in which only nine sentences described the presentation of their own groups and the rest mentioned other groups' presentations.

For problem-solving and finding, the word "data" was utilized in relation to "collection or collect" 139 times, "comparison" 91 times, and "analysis" 49 times. Another notable finding is that the word "data" was utilized along with the word "Japan" 25 times and the words "study or examine" 28 times. The word "problem" was utilized along with the word "solve" 42 times, and all sentences that utilized "problem" were listed in relation to the selected topics of the countries that students chose for proposal making for solving issues. Seven sentences were in relation to understanding problems by collecting relevant data. The word "interview" appeared 115 times and stands in co-occurrence analysis. The word "interview" was utilized with the word "online" 25 times and with "on-site" 19 times.

For multicultural understanding and communication, the word "group" was utilized in the context of "group work" 217 times, "group members" 78 times, and "group leader" 18 times. In sentences including the word "group" and "leader," 17 sentences were about actual work as a leader, while one sentence was about the relationship with a leader. The word "English" was utilized 42 times, in the context of "communication, talk and

conservation" 39 times and with "TA" 24 times. The word "TA" was utilized in relation to the word "international students" 24 times, "communication" 31 times, "group" 25 times, "English" 24 times, "members" 17 times, and "interview" 10 times, respectively. The word "interview" appeared 63 times and was utilized in the context of "online interview" 10 times and with "TA" 10 times as well.

**Table 3.** Context utilization of frequently appeared words in each skill category. (Original data collected from submitted essays from students registered for "Introductory Course for Global Scientists and Engineers" in February 2021).

| Skill Category | Word | Frequency | Utilization within the Sentences |
|---|---|---|---|
| **Global Awareness** | Japan | 77 | Comparing to other countries |
| | | 89 | with the word "country" |
| | | 39 | with the word "problem" |
| | | 24 | with the word "study" |
| | | 22 | with the word "similarities and differences" |
| | | 21 | with the word "other countries" |
| | Problem | 61 | other countries' problems |
| | | 53 | selected countries' problems |
| | | 59 | relations to field of expertise |
| | | 89 | with the word "solve" |
| | | 27 | with the word "environment" |
| | | 25 | with the word "society" |
| | Country | 89 | with the word "Japan" |
| | | 46 | selected countries |
| **Problem solving and finding** | Presentation | 89 | other groups' presentations |
| | data | 139 | with the word "collection or collect" |
| | | 91 | with the word "comparison" |
| | | 49 | with the word "analysis" |
| | | 28 | with the word "study or examine" |
| | | 25 | with the word "Japan" |
| | Problem | 42 | with the word "solve" |
| | | all | in the context of selected countries |
| | | 7 | in the context of collected data |
| | Interview | 25 | with the word "online" |
| | | 19 | with the word "on-site" |
| **Multicultural understanding** | group | 217 | group work |
| | | 78 | group members |
| | | 18 | group leaders |
| | English | 42 | in the context of communication |
| | | 39 | in the context of talk and conversation |
| | | 24 | with the word "TA" |
| | TA | 24 | with the word "international students" |
| | | 31 | with the word "communication" |
| | | 25 | with the word "group" |
| | | 17 | with the word "member" |
| | | 10 | with the word "interview" |
| | Interview | 10 | in the contect of online interview |
| | | 10 | with the word "TA" |

### 5.4. Result of Content Analysis

Table 4 shows the inclusion of data and online interviews for the final report submitted by each group. Students selected many of the 17 themes of SDGs, except three themes, which are "affordable and clean energy," "peace and justice and strong institutions," and "partnerships for the goals". The most selected SDG was "sustainable cities and communities," of which 11 groups were chosen. "Gender equality" and "climate action" were selected by five groups, followed by "good health and well-being" and "quality education" picked by four groups. Three groups selected "no poverty." Two groups selected" life below water" and "decent work and economic growth". The themes "zero huger," "industry, innovation, and infrastructure," "reduced inequality," and "live on land" were selected by one group each. Of 45 groups, 41 groups searched for additional items relevant to their selected theme, and one-third (15 groups) of the groups added more than five items to their data collection. All 45 groups included data and interview to understand the current situation of the selected topic. 16 groups utilized collected data for proposal making, while most of the groups—44 groups—utilized the interview results for proposal making. 15 groups included both data and interviews to proposal making.

**Table 4.** Selected themes in relation to SDGs' goal and utilization of data and interview to the final report (Original data collected from submitted reports from students registered for "Introductory Course for Global Scientists and Engineers" in February 2021).

| SDGs | | Topic | Country | Data | | | Interview | |
|---|---|---|---|---|---|---|---|---|
| No. | Themes | | | No. of Additional Item | Use for Current Situation | Use for Proposal | Use for Current Situation | Use for Proposal |
| 1 | No poverty | Pollution Free Revolution in India | India | 3 | ✓ | | ✓ | ✓ |
| | | Poverty in India | India | 0 | ✓ | | ✓ | ✓ |
| | | Street Children in Vietnam | Vietnam | 4 | ✓ | | ✓ | ✓ |
| 2 | Zero hunger | Reducing economic disparities by improving agriculture | China | 2 | ✓ | ✓ | ✓ | ✓ |
| 3 | Good health and well-being | Well-being after Retirement in Bangladesh | Bangladesh | 2 | ✓ | | ✓ | ✓ |
| | | Improving primalry medical system in rural area | India | 10 | ✓ | ✓ | ✓ | ✓ |
| | | Medical care in Indonesian remote island | Indonesia | 7 | ✓ | | ✓ | ✓ |
| | | Minimize the negative influence of aging society | Philippines | 4 | ✓ | | ✓ | ✓ |
| 4 | Quality education | Educational gap and employment rate in China | China | 6 | ✓ | | ✓ | ✓ |
| | | Education and slam | Indonesia | 13 | ✓ | | ✓ | ✓ |
| | | Educational Inequality in Mexico | Mexico | 4 | ✓ | ✓ | ✓ | ✓ |
| | | Education gap in Thailand | Thailand | 4 | ✓ | | ✓ | ✓ |
| 5 | Gender equality | Child marriage in Bangladesh | Bangladesh | 3 | ✓ | ✓ | ✓ | ✓ |
| | | Child marriage in Bangladesh | Bangladesh | 8 | ✓ | | ✓ | |
| | | Women empowerment in Bangladesh | Bangladesh | 7 | ✓ | ✓ | ✓ | ✓ |
| | | Property inheritance problem of Hindu women in Bangladesh | Bangladesh | 2 | ✓ | | ✓ | ✓ |
| | | Gender Inequality | Vietnam | 7 | ✓ | ✓ | ✓ | ✓ |
| 6 | Clean water and sanitation | Water Crisis in India | India | 1 | ✓ | ✓ | ✓ | ✓ |
| | | Arsenic Water Contamination and Treatment in Eastern India | India | 3 | ✓ | | ✓ | ✓ |
| 7 | Decent work and economic growth | Make ideas of solving environmental issues in Galapagos with balancing environment and economy(=tourism) | Ecuador | 2 | ✓ | | ✓ | ✓ |
| | | Unemployment in India | India | 4 | ✓ | | ✓ | ✓ |
| 8 | Industry, innovation and infrastructure | Winter Tourism in Austria | Austria | 4 | ✓ | | ✓ | ✓ |
| 9 | Reduced inequalities | How to solve the problems led by the rapid population growth in India? | India | 3 | ✓ | | ✓ | ✓ |

Table 4. *Cont.*

| No. | Themes | Topic | Country | No. of Additional Item | Data Use for Current Situation | Use for Proposal | Interview Use for Current Situation | Use for Proposal |
|---|---|---|---|---|---|---|---|---|
| 10 | Sustainable cities and communities | Traffic congestion in Bangladesh | Bangladesh | 2 | ✓ | | ✓ | ✓ |
| | | Food Waste Management in Dhaka City | Bangladesh | 1 | ✓ | | ✓ | ✓ |
| | | Study on air pollution from car exhaust in Cairo | Egypt | 6 | ✓ | | ✓ | ✓ |
| | | Improvig road planning | Egypt | 9 | ✓ | | ✓ | ✓ |
| | | The Construction of Steel Flyover-Solution To The Traffic Congestion in Bangalore, India | India | 0 | ✓ | | ✓ | ✓ |
| | | Reducing air pollution from open burning and firecracker during Diwali | India | 3 | ✓ | | ✓ | ✓ |
| | | How to reduce the number of illegal dumping | Malaysia | 8 | ✓ | | ✓ | ✓ |
| | | Land transport in Metro Manila, | Philippines | 8 | ✓ | ✓ | ✓ | ✓ |
| | | Improving transportation system | Vietnam | 2 | ✓ | ✓ | ✓ | ✓ |
| | | surplus motorbike & traffic congestion | Vietnam | 3 | ✓ | | ✓ | ✓ |
| | | Transport infrastructure in Vietnam | Vietnam | 9 | ✓ | ✓ | ✓ | ✓ |
| | | Reducing Air Pollution Caused by Motorcycles | Vietnam | 0 | ✓ | | ✓ | ✓ |
| 11 | Responsible consumption and production | Waste Management in Shanghai and Our Proposal for the Secondhand Clothing Market | China | 9 | ✓ | | ✓ | ✓ |
| | | How to solve Food Safety in Vietnam | Vietnam | 3 | ✓ | | ✓ | ✓ |
| 12 | Climate action | How to encourage Bangladeshis in coastal area to evacuate during cyclone | Bangladesh | 2 | ✓ | ✓ | ✓ | ✓ |
| | | Proposing Solutions to Some Aspects of Flooding in the Philippines | Philippines | 1 | ✓ | ✓ | ✓ | ✓ |
| | | Flood prevention in Philippines | Philippines | 6 | ✓ | ✓ | ✓ | ✓ |
| | | Countermeasures for flood in Thailand | Thailand | 0 | ✓ | ✓ | ✓ | ✓ |
| | | Flooding in Ayutthaya | Thailand | 5 | ✓ | ✓ | ✓ | ✓ |
| 13 | Life below water | Increasing fish consumption in Indonesia | Indonesia | 7 | ✓ | | ✓ | ✓ |
| | | Increasing fish consumption in Indonesia | Indonesia | 3 | ✓ | | ✓ | ✓ |
| 14 | Life on land | Biodiversity of Malaysia | Malaysia | 8 | ✓ | ✓ | ✓ | ✓ |

### 5.5. Summary of Research Results

From the research results, the following findings are summarized. First, most of the topics listed on SDGs are covered in this online PBL—specifically, 45 selected topics related to 14 themes among 17 SDGs. Many groups chose topics related to sustainable cities and communities, such as traffic jams, waste management, air pollution, water management, and so forth. Second, global awareness grew among students in relation to Japan and the consideration of similarities and differences with selected countries. By conducting research on problems in the selected country through group work and in trying to find solutions, students deepened their knowledge of their target country and selected issues. Many students tried to refer to their selected issue along with their field of expertise. As many students referred to the word "presentation" as other groups' presentations, and many sentences in their submitted essays indicated other countries, many of the students learned about issues in other countries by listening to the other groups' presentations in the class. Thus, "problem" in the context of a skill category on global awareness refers to both problems that students researched on their own and also problems that other groups studied. Third, as the data and interview appeared frequently in all three text mining analyses, it is assumed that problem finding and solving skills were developed through data collection and comparative analysis between selected countries and Japan as well as online interviews with the person from the targeted countries. Within the essay related to the skill category problem finding and solving, the word "problem" frequently used in the

essays refers to selected issues that students worked on for their own group work, which is different from how it was used in essays on global awareness. Fourth, multicultural understanding and communication were facilitated through interactions with both the TA and group members. English communication was utilized in communication with international students as TAs as well as by conducting interviews with a person living in the targeted country.

### 6. Reasons for Research Findings

From the aforementioned research results, it is concluded that through online PBL with international students as TAs, students deepened knowledge about issues related to SDGs and developed all three skill sets listed as learning objectives of the course, namely global awareness, problem finding and solving, and multicultural understanding and communication. Specific reasons for each research finding are explained as follows.

First, in increasing global awareness in relation to SDGs, students considered problems of targeted countries in relation to Japan and tried to find solutions. Topics on SDGs were well covered through this PBL course because of TAs' suggestion and assistance; that is to say, at the beginning of the group work, each TA made a presentation to the group members that he/she was assigned to assist and presented three possible topics that group members would work on. Group members then selected topics of their interest from different perspectives such as their interest, emergency of the issues, the relation to their field of expertise, close link with their own problems and such. Selections of some topics related to SDGs are linked with students' fields of expertise and interests. That is to say, many groups have chosen the topics either in a way that can bring interest to prospective engineers or students who are receiving education.

Second, a remarkable number of students tried to consider ways to make a contribution to selected issues because one of the missions for Tokyo Tech is to grow human resources to deal with complicated issues in the world. Students were trained and asked on many occasions such as their university entrance ceremony and introduction of the online PBL to consider the roles of engineers for the establishment of a sustainable society. Thus, students are often trying to consider within their career plan to extend their future for the well-being of society.

Third, it is assumed that students have enhanced understanding and interest in other countries' issues through listing to other group presentations. Efforts to carefully listen to other groups' presentations were made in a way that students were asked to make comments in a few sentences for each presentation. Comments were then submitted as an assignment. Finally, instructors shorted, compiled, and shared them with each group. Students were therefore able to pay good attention to other groups' presentations and compare research results and proposals with their own studies.

Fourth, for problem finding and solving, it was assumed that the assignment on data collection and comparative analysis was utilized mainly for three purposes: to understand overall situations of the selected countries with the comparison of that of Japan, to picture overall situations of the selected topics, and to make a direction of proposal. Furthermore, through the online interview, students could grasp real situations of the topics in selected countries as first-hand information provided by local people, which could not be obtained from secondary information and data otherwise. Through online interviews, students could study the real insight of the countries about selected topics. Through data analyses and online interviews, students were able to conduct both objective and subjective analyses of selected topics and come up with feasible proposals acceptable to their target countries.

Fifth, multicultural understanding and communication skills have been developed through the whole process of group work as well as discussion with international students as TAs. It was assumed that students communicated with TAs in English, while overall communication skills such as coordination, discussion, acceptance, sharing were developed with group members. Prior to starting the very first online group work, instructors briefed manners of online group work. Some tips included securing multiple channels of

communication, putting videos on as much as possible, starting with a self-introduction, using reaction icons, the chat function and so forth, refraining from denying other people's opinions, using negative words and avoiding personal criticism, taking turns in making opinions, and confirming next action steps.

## 7. Discussion

This research focused on students' learning outcomes and processes of online PBL conducted during COVID-19 in the 2020 academic year. In this section, based on the aforementioned research findings and by assessing merits and demerits of online PBL, pedagogies to ensure effective learning equivalent to traditional face-to-face PBL are discussed.

Merits of online PBL are closely linked with the effective utilization of ICT tools. For knowledge sharing and acquisition of selected issues, it can be well ensured by a combination of asynchronous and synchronous online learning. Specifically, on-demand videos, learning materials, and recorded lectures that are uploaded on the LMS can be utilized to re-confirm the learning contents. Students can concentrate on listening to instructors' lectures through online learning than classroom learning because student sits alone in their own room to focus on talks without distraction. Moreover, documents and information can be easily shared and edited with shared drive creation, while discussion and internet searches can be done simultaneously during online group work. They can focus on the group work, and it is easier to talk online if group members take turns in expressing their opinions. Furthermore, it is easy to adjust group meetings outside of the class because time is saved to come to physically meet in one place, so students can meet online even in the late evening. Meeting people without considering actual distance was one of the great benefits of conducting online interviews with people living in countries outside of Japan. Without any cost for international transportation, students were able to meet and ask questions about issues in their target countries.

Demerits of online learning include that compared to face-to-face PBL, it is difficult to develop initial communication to establish friendly relationships and building trust among group members. Online group work, as it only utilizes sounds and visuals in a set frame, takes longer to understand the feelings of other group members. Additionally, if the timing is lost, it is difficult to express comments or opinions to other group members. Discussion and networking are only saved when students are connected online; thus, they cannot share any other moments before and after the set time length for the online discussion or formal class, which makes it a little difficult to ask small questions that can be easily confirmed with traditional face-to-face learning.

Other pedagogies such as group work on selected issues with proper task sharing, asking students to write comments to other group presentations, putting assignments related to data collection, comparison and analysis, and multicultural communication with international students as TAs in English can be implemented in both online and face-to-face PBL courses. Thus, by maintaining the merits of online PBL, while overcoming the demerits, which is mostly related to communication methods, it is concluded that effective pedagogy with proper students–teacher–TA interaction and follow-up can be well delivered as state-of-the-art teaching even after the COVID-19 pandemic. One of the characteristics of online PBL, especially during the pandemic, is that in addition to the general restriction of mobility in everyday life, students, TAs, and instructors are physically separated; thus, many efforts to bring lots of communication during group work and with the whole course delivery should be implemented. Ways to reach students over the screen within the whole course period should be carefully considered and determined. It is sometimes stressful to talk in front of one's computer alone. Communication can also be well delivered if instructors ask to put the video on, and if instructors, TAs, or group leaders can facilitate the initial group work each time. In this regard, the groupwork report submitted by each group and its feedback by instructors was well operated to ensure the progress and track any problems involved during online PBL.

## 8. Conclusions

Considering the experiences authors faced during the COVID-19 pandemic, it is assumed that "global education" can be implemented anywhere and anytime. That is to say, when considering the words "global education," many people used to first imagine a study abroad which requires physically crossing the country's border. While COVID-19 certainly accelerated the shift to online PBL, just like many workers have shifted their lifestyle to work at home or anywhere else but traditional office settings, the ability to collaborate and communicate online across countries and regions is becoming an increasingly important aspect for the global workplace. In realizing online PBL as part of new pedagogy for global education anywhere and anytime, the following points should be noted. First, the utilization of online tools is necessary to promote online PBL as part of global education without mobility. Second, in implementing online PBL, instructors should pay good attention to each student even more than in traditional face-to-face learning. Third, there are many learning methods that can be provided through online PBL, such as content delivery, group discussion, online interview, and others, all of which need some technical adaptation, especially for instructors to organize the course. Instructors therefore should keep strong attention in implementing new tools for effective online PBL delivery. Fourth, there are certain experiences/learning that cannot be obtained through online PBL as part of global education. In that case, other means such as studying abroad or face-to-face interaction should be promoted. Fifth, instructors should pay good attention to students' comments and suggestions for the effective organization of online PBL and global learning for further improvement of courses. As many pieces of literature insist, with the aforementioned course delivery in this research, authors strongly believe that online PBL with the component of multicultural understanding and communication can be one of the effective pedagogies for engineering education.

**Author Contributions:** Conceptualization, E.O. and R.M.-S.; methodology, E.O. and R.M.-S.; formal analysis, E.O. and R.M.-S.; investigation, E.O. and R.M.-S.; resources, E.O.; data curation, E.O. and R.M.-S.; writing—original draft preparation, E.O.; writing—review and editing, E.O.; visualization, E.O. All authors have read and agreed to the published version of the manuscript.

**Funding:** The organization of the on-line PBL was funded by Ministry of Education, Culture, Sports, Science and Technology in Japan, under "The Top Global University" project.

**Institutional Review Board Statement:** Not applicable.

**Informed Consent Statement:** Not applicable.

**Data Availability Statement:** Data is available upon request.

**Acknowledgments:** Authors would like to express sincere appreciation to international students at Tokyo Tech, who served as TAs for the online PBL course, to bring diversity and introduce issues in their own countries. Special thanks also goes to students who registered for the PBL course during difficult time of COVID-19 pandemic.

**Conflicts of Interest:** The authors declare no conflict of interest.

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
