# Peer review of "Effects of Online Problem-Based Learning to Increase Global Competencies for First-Year Undergraduate Students Majoring in Science and Engineering in Japan"

_sustainability, doi:10.3390/su14052988_

Round 1

Reviewer 1 Report

The topic is really interesting and relevant. In general, the elaboration and the applied methodology is accurate.

The bibliographic review could still be broadened as the international bibliography is really rich in the field, and even in MDPI-Sustainability similar studies have been recently published, for example, Madlenak et al (2021) https://www.mdpi.com/2071-1050/13/14/7675/pdf

Apart from this, the paper from substantial point of view is very good. However, there are some formal problems due to which the paper also needs minor corrections.

The sources of data and information are missing in case of ALL tables and figures (they should be put under the figures and tables). It is absolutely a formal requirement in case of academic papers, even if the data were gained by the direct (primary) research of the authors.

Another problem is that in tables the font style is different from the body text, it should be identical. Some of the tables and figures – which were probably copy-pasted from other papers are not well visible, the letters’ contour seems to fade or the text is too small, it is difficult to read them, so re-editing of such figures, possibly all of them, but at least figure 1 is highly recommended.

 The conclusion part should NOT contain quotations from other authors (like Wojenski) as in this chapter the authors should discuss ONLY their own results and specify suggestions of applicability - possibly in a structured form – upon their findings. Quotations, citations can be done in the previous chapters especially in the literature review part.

After the mentioned minor improvements were done, the paper can be recommended to be accepted and published.

Author Response

Dear Reviewer, 

Thank you very much for sending us the valuable comments.  It was very effective to improve our article.  Based on your comments, we have further revised our article as follows. 

1) We have added 5 more bibliography related to on-line learning, especially during COVID-19 pandemic.

2) We have dully changed the font identical to body text, for all  tables. 

3) Figure 1 and all other figures have been replaced with high visibility.

4) We have reorganized the quotations in conclusion and included in literature review.

Truly appreciate your assistance, we shall look forward to your final decision. 

With our best and warmest regards, 

Authors

Reviewer 2 Report

The authors' study makes a good impression: the article has a clear logical structure, the methodology is well described, the conclusions and points of discussion are described in detail and described for the study itself. Sufficient base of the studied theoretical materials.

However, I have 3 remarks:

  1. The article requires technical verification: there are many extra spaces after the dots at the end of the sentence.

2. Need to redraw Figure 1 - small unreadable text

3. The purpose of the study and/or hypothesis should be more clearly stated.

Author Response

Dear Reviewer, 

Thank you very much for sending us the valuable comments.  It was very effective to improve our article.  Based on your comments, we have further revised our article as follows. 

1) We have dully reduced spaces in each sentence to one space only.

2) Figure 1 and all other figures have been replaced with high visibility.

3) We have re-stated the purpose of this study.  

Truly appreciate your assistance, we shall look forward to your final decision. 

With our best and warmest regards, 

Authors